# Extended Cleavage Specificity of two Hematopoietic Serine Proteases from a Ray-Finned Fish, the Spotted Gar (*Lepisosteus oculatus*)

**DOI:** 10.3390/ijms25031669

**Published:** 2024-01-30

**Authors:** Paolo Valentini, Srinivas Akula, Abigail Alvarado-Vazquez, Jenny Hallgren, Zhirong Fu, Brett Racicot, Ingo Braasch, Michael Thorpe, Lars Hellman

**Affiliations:** 1Department of Cell and Molecular Biology, Uppsala University, P.O. Box 596, SE-751 24 Uppsala, Sweden; paolovalentini94@gmail.com (P.V.); srinivas.akula@icm.uu.se (S.A.); annazhirongfu@gmail.com (Z.F.); michael.thorpe@thermofisher.com (M.T.); 2Department of Medical Biochemistry and Microbiology, Uppsala University Biomedical Centre (BMC), P.O. Box 582, SE-751 23 Uppsala, Sweden; abigail.vazquez@imbim.uu.se (A.A.-V.); jenny.hallgren@imbim.uu.se (J.H.); 3Department of Integrative Biology, Michigan State University, East Lansing, MI 48825, USA; racicotb@msu.edu (B.R.); braasch@msu.edu (I.B.); 4Ecology, Evolution and Behavior Program, Michigan State University, East Lansing, MI 48825, USA

**Keywords:** fish, serine protease, cleavage specificity, tryptase, macrophage, evolution

## Abstract

The extended cleavage specificities of two hematopoietic serine proteases originating from the ray-finned fish, the spotted gar (*Lepisosteus oculatus*), have been characterized using substrate phage display. The preference for particular amino acids at and surrounding the cleavage site was further validated using a panel of recombinant substrates. For one of the enzymes, the gar granzyme G, a strict preference for the aromatic amino acid Tyr was observed at the cleavable P1 position. Using a set of recombinant substrates showed that the gar granzyme G had a high selectivity for Tyr but a lower activity for cleaving after Phe but not after Trp. Instead, the second enzyme, gar DDN1, showed a high preference for Leu in the P1 position of substrates. This latter enzyme also showed a high preference for Pro in the P2 position and Arg in both P4 and P5 positions. The selectivity for the two Arg residues in positions P4 and P5 suggests a highly specific substrate selectivity of this enzyme. The screening of the gar proteome with the consensus sequences obtained by substrate phage display for these two proteases resulted in a very diverse set of potential targets. Due to this diversity, a clear candidate for a specific immune function of these two enzymes cannot yet be identified. Antisera developed against the recombinant gar enzymes were used to study their tissue distribution. Tissue sections from juvenile fish showed the expression of both proteases in cells in Peyer’s patch-like structures in the intestinal region, indicating they may be expressed in T or NK cells. However, due to the lack of antibodies to specific surface markers in the gar, it has not been possible to specify the exact cellular origin. A marked difference in abundance was observed for the two proteases where gar DDN1 was expressed at higher levels than gar granzyme G. However, both appear to be expressed in the same or similar cells, having a lymphocyte-like appearance.

## 1. Introduction

Serine proteases constitute the major granule content of several hematopoietic cell types. Large amounts of active chymotryptic and tryptic enzymes are stored in the cytoplasmic granules of mammalian mast cells for rapid release upon activation. Instead, mammalian neutrophils store primarily elastolytic enzymes, as exemplified by neutrophil elastase and proteinase 3, but also a tryptic enzyme, NSP-4, and a chymotryptic enzyme, cathepsin G [1,2,3,4]. Human cytotoxic T cells and natural killer cells (NK cells) express and store several granzymes with tryptase, asp-ase and met-ase specificities, as exemplified by granzymes A, B, and M, respectively [1]. A relatively detailed picture has been established of how these enzymes have appeared and diversified during mammalian evolution [2,3,5,6,7,8,9,10,11,12,13,14,15]. However, the information concerning the presence and specificity of such enzymes in reptiles, amphibians, and fishes is still only fragmentary [16,17,18,19,20,21,22,23]. There has been an increase in the use of non-rodent animal models with much focus on non-mammalian vertebrates including fish species, notably the zebrafish. However, there are few studies on the details regarding the characterization of hematopoietic serine proteases in other fishes. Here, the (spotted) gar is of particular interest as it represents a basally diverging branch of the ray-finned fish evolutionary tree and has been found to have a genome that has undergone a relatively limited number of rearrangements and amplifications compared to many other fish lineages including the zebrafish, which is often used as a model organism [24]. The gar genome thereby represents a less derived configuration than most other fish genomes and has interestingly been found to show major similarities with mammalian genomes [24]. All vertebrates have experienced at least two tetraploidizations. The first was an early event that has been named 1R, that recently has been estimated to have taken place during early Cambrian 535.3–524.8 Ma, before the divergence of crown vertebrates that has been dated to 510–506 Ma [25]. The second is the 2R, which occurred before the divergence of the crown gnathostomes (the jawed vertebrates) and that has been dated to late Cambrian and early Ordovician, 498.4–485.2 Ma [25]. Teleosts, but not bichir and gar, have experienced at least one additional tetraploidization, the 3R or Teleost Genome Duplication (TGD), which has been dated to approximately 320 Ma [26]. The sterlet sturgeon has undergone its own, separate third round of whole genome duplication at approximately 180 Ma [27]. In addition, several teleost lineages have experienced additional and independent tetraploidizations, including salmonids and carp, which have added a lot of new genetic material by genome doublings that have resulted in accelerated evolutionary rate of protein-coding and nucleotide sequences, a higher rate of intron turnover, a loss of many cis-regulatory elements, and shorter conserved syntenic blocks [28]. These genomes are therefore highly derived, and it is often difficult to define corresponding genomic regions between fish and mammalian genomes [28]. In order to look closer into the early events in the expansion and diversification of these hematopoietic serine proteases, this analysis therefore focuses on two hematopoietic serine proteases from one of the fish species that has experienced only two whole genome duplications like humans and other mammals and thereby less dramatic rearrangements of their original genome, the spotted gar [24].

In mammals, serine proteases play important and diverse roles in a number of physiological processes including blood coagulation, food digestion, fertilization, immunity, and tissue repair [29]. This large chymotrypsin/trypsin family all share a common mechanism for cleaving peptide bonds, based on their catalytic triad, with three vital residues: His57, Asp102, and Ser195 (chymotrypsinogen numbering) [30] (Figure 1A). These key amino acids are located near a substrate-binding pocket (termed S1), typically made up of residues 189, 216, and 226 (also chymotrypsinogen numbering) [30] (Figure 1B). Together, they form the specificity-conferring triplet and provide clues on the primary specificity of the serine proteases (Figure 1B). For mammalian enzymes and also other tetrapods, these three residues provide a relatively good indication for the primary specificity of an enzyme. However, for the majority of fish proteases, excluding the granzyme A/K members, these three residues provide little guidance to their primary specificities. This is primarily due to large sequence divergences, making the positioning of the relevant residues difficult to determine based only on the primary sequence. To determine the specificity of the fish proteases, we therefore have to rely on direct experimental analysis.

The primary specificity of a protease is the amino acid after which the cleavage occurs, and this residue is named the P1 residue (Figure 1A). The residues present at the N-terminal of the cleavage site are numbered P2, P3, P4, etc., and the residues present at the C-terminal of the cleavage site are numbered P1′, P2′, P3′, etc. (Figure 1A). The primary specificity determines the main specificity of the enzyme. If the enzyme cleaves after a basic amino acid, such as Arg or Lys, it has tryptic activity, and when it cleaves after large aromatic amino acids, such as Phe, Tyr, or Trp, it has a chymotryptic activity (Figure 1B). The extended specificity is determined by the selectivity in residues surrounding the cleavage site, most often the P2, P3, P4, P1′, P2′, and P3′ residues (Figure 1A).

To determine the cleavage specificity of an enzyme, different techniques can be used, including chromogenic substrates, peptide libraries, and phage display. Substrate phage display is a method that can provide the detailed information concerning both the primary and extended specificities of an enzyme. This method has been used to study the extended specificities of two serine proteases from the spotted gar. These two proteases were extracted from the genomic database depending on the distance of the relationship with other known hematopoietic serine proteases. Both enzymes showed very stringent primary and extended specificities, with one being a Tyr-specific chymase and the other being a Leu-specific Leu-ase. Antisera developed against these two proteases detected cells in the Peyer’s patch-like structures of the intestinal region of the gar, with a pattern of expression indicating that the proteases are expressed in cytotoxic T cells or NK cells. However, it is not yet possible to more specifically identify the cell origin as very few reagents are available for studies of immune cells in the gar.

## 2. Results

### 2.1. Phylogenetic Analyses

Human and mouse hematopoietic serine protease sequences were used as query sequences to identify similar sequences in a large panel of vertebrate genomes in the NCBI database using the TBLASTN algorithm. The alignment, using MAFFT and the MrBayes program, generated a Bayesian phylogenetic tree that is depicted in Figure 2A. An enlarged version of the fish proteases clustering in a separate branch of the tree is shown in Figure 2B. The phylogenetic analyses were performed essentially as described in a previous publication using the same strategy and sequences [2].

### 2.2. Production, Purification, and Activation of Gar Granzyme G and Gar Duodenase 1

DNA constructs containing the coding regions for the gar granzyme G (Gzm-G) and the gar duodenase 1 (DDN-1), an N-terminal His_6_-tag, and followed by an enterokinase (EK) site were designed and ordered from Genscript. These fragments were subsequently cloned into the mammalian expression vector pCEP-Pu2 for expression in HEK 293-EBNA cells [33]. Here, the His_6_-tag facilitates the purification with Ni^2+^ chelating immobilized metal ion affinity chromatography columns, and the cleavage with EK activates the enzyme, whilst simultaneously removing the His_6_-tag and the EK site (Figure 3).

### 2.3. Substrate Phage Display

To determine the extended cleavage specificity of the two gar enzymes, a phage T7 based system was used where the individual peptide sequences are displayed on the surface of the phage. This system enables the characterization of a region covering both 4-5 amino acids N-terminally and the cleavage site C-terminally. The library used had a complexity of approximately 50 million different peptide sequences. After 7 selection rounds, the gar enzymes selected phages, which showed at least a three order of magnitude increase compared to the PBS control. One hundred and twenty phage plaques from the last selection round were picked from both of these proteases, the region encoding the peptide sequence was amplified using PCR, and the 96 clones with the best quality PCR bands from both of them were sent for sequencing. Following the decoding of the DNA sequence, the amino acid sequence of the variable linker region was aligned. Each row represents an individually sequenced random region, and multiple similar sequences are shown to the right where necessary (Figure 4).

The alignment showed a highly specific selection, with an apparent preference for Tyr in the P1 position for gar granzyme G, indicating a highly selective chymase activity (Figure 4). There were also fairly strict preferences in the extended specificity: aromatic amino acids or Pro in the P2 position, Val in the P3 position, Ala or Gly in P4, Ser or Met in P1′, no negatively charged residues in P2′, P3′, andP4′ positions, Leu or Gly in the P2′ position, and a slight preference for Val in the P3′ position (Figure 4).

The gar DDN1 was found to be a strict Leu-ase with a high preference for Leu in the P1 position, Pro in the P2 position, Leu in the P3 position, and Arg in both the P4 and P5 positions (Figure 4). A preference for Leu, Ser, or Met in the P1′ position and a lower selectivity in both the P3′ and P4′ positions (Figure 4).

### 2.4. Phage Display Sequence Verification Using Recombinant Substrates

In order to validate the phage display sequence data and to address small variations of amino acids in the aligned phages, a system has been developed in our lab whereby a number of sequences were analyzed by the cleavage of recombinant substrates in a two-trx system (Figure 5). This is based on the expression of the selected cleavable amino acid sequences derived from the phage display data placed in a linker region between two *E. coli* trx molecules. Various amino acids within the sequences were changed to pinpoint the efficiency and selectivity of the two gar enzymes (Figure 5A,B). The analysis of the selectivity in the P1 position for gar granzyme G showed a 3-5-fold higher preference for Tyr over Phe in the P1 position and that the enzyme did not tolerate another aromatic amino acid, Trp, in this position (Figure 5C). In the P1′position, the negatively charged amino acid Glu could replace the Ser of the consensus sequence without a decrease in cleavage activity (Figure 5C). Phe could also replace the most preferred Pro in the P2 position without any change in cleavage activity and also an Arg could replace the Val in the P3′position (Figure 5D). However, introducing a larger amino acid, in this case, a Trp in the P3 position instead of the Val of the consensus, resulted in an approximately 10-fold reduction in cleavage (Figure 5D).

When analyzing the gar DDN1, exchanging the Leu for a Ser in the P1 position completely blocked cleavage, which was expected based on the phage display result (Figure 6A). However, only a minor reduction in cleavage activity was seen when changing the Pro in the P2 position with an Ala (Figure 6A). A more pronounced effect was seen when changing the Pro with an Arg in the P2 position, resulting in an approximately 3–5-fold reduction in cleavage efficiency (Figure 6A). Changing the P2 Pro with a Leu also resulted in a marked reduction in cleavage (Figure 6C, second sequence). Interestingly, changing the P1′ Leu for a Met enhanced cleavage quite substantially, almost tenfold (Figure 6B). This phenomenon is rarely seen as the phage display usually provides the most preferred cleavage site. However, Leu is encoded by six different codons and Met with only one, and Leu is therefore much more frequently occurring in a random sequence, which may be the reason for this relatively high number of Leu in the P1′ position in the phage display, although we also found a significant number of Met in this position in the phage display sequences (Figure 4). The Arg residues in positions P4 and P5 appear to be very important for cleavage as a very marked reduction occurred by changing P4 Arg with a Gly (Figure 6B). Exchanging both Arg P4 and Arg P5 with Gly resulted in an almost total block in cleavage (Figure 6C). The positions of these two Arg residues in P4 and P5 also seemed to be very important as seen from the first sequence in Figure 6C.

### 2.5. Screening for Potential In Vivo Substrates

The initial screening was confined to the gar proteome with the consensus sequences (Ala-Val-Phe-Tyr-Ser-Leu) and (Arg-Arg-Leu-Pro-Leu-Leu) for gar granzyme G and DDN1, respectively. Both of these sequences provided a number of hits in the gar proteome. However, there is no clear indication to what the potential function that could be deduced with these potential target proteins. The AVFYSL consensus sequence obtained for gar granzyme G resulted in a number of extracellular proteins and cell surface receptors including protocadherin Fat 1 isoforms, neogenin isoforms, SID-1 transmembrane family members, D3 dopamine receptor isoforms, platelet-activating receptor isoforms, and the sodium-dependent noradrenalin and the glycin transporter isoforms. The RRLPLL consensus sequence obtained for gar DDN1 resulted in another diverse set of potential target proteins including diphamide biosynthesis protein 2 isoforms, tyrosine-protein kinase BAZ1B, adenylate cyclase type 10, RNA-binding protein 10-like, tubulin polyglutamylase TTLL5, caspase recruitment domain-containing protein 11, E3 ubiquitin-protein ligase RNF216, sodium bicarbonate transporter-like protein 11 isoforms, zinc finger homeobox 4 isoforms, kelch-like protein-1 isoforms, and additional targets.

### 2.6. Analyzing Gar Tissue for Sites of Expression for the Two Gar Enzymes

To look at the tissue distribution of these two gar enzymes, gar tissue was required. Therefore, the upper part of the body of five juvenile gars were obtained. Tissue sections originating from the region of the fish marked by a red arrow in the tissue are stained with hematoxylin–eosin (Figure 7B). The typical villi of the intestinal wall are seen, and a white arrowhead marks a Peyer’s patch-like structure in the intestinal wall (Figure 7C). An enlargement of this region is shown in Figure 7D. In Figure 7E,F, a section of the head kidney, a hematologic organ in fishes, was analyzed. In this organ, a marked division in separate regions can be observed. There were regions with almost 100% red blood cells and other regions where an heterogenous mix of different cells, most likely immune cells, were observed. In these regions, there were only a few red blood cells, which seem to come in small clusters, possibly in blood vessels (Figure 7E,F).

Tissue sections from the same region were used for immunohistochemical analyses with two rat polyclonal antibodies directed against the two gar enzymes. As seen in Figure 8A,B, a number of positive cells in the Peyer’s patch-like region were observed for gar DDN1. Much lower intensity signals and fewer positive-stained cells for the gar granzyme G antisera were observed in the same Peyer’s patch region (Figure 8C). White arrowheads mark positive cells in Figure 8A–C. No staining was seen with a pre-immune serum (Figure 8D).

### 2.7. Gene Loci for Gar Hematopoietic Serine Proteases

Both of the analyzed gar enzymes were located in the met-ase locus, and the locus in humans encodes the majority of the neutrophil proteases, including N-elastase (ELANE), proteinase 3 (PRTN3), neutrophil serine protease 4 (NSP-4 or PRSS57), and the inactive azurocidin (AZU1), and also complement factor D (CFD) and granzyme M (GZMM) (Figure 9). Interestingly, only one enzyme, the gar DDN1, was found in the region orthologous to the human locus where four serine proteases were found including CFD, ELANE, PRTN3, and AZU1, whereas five protease genes were found in a region of the locus where only one gene was found in the human locus the PRSS57 (NSP4) (Figure 9). This indicated that separate gene duplication events have resulted in an increase in serine protease numbers in this locus in fish and tetrapods. An inversion also seems to have occurred most likely in the human locus as all three early branches of fishes have the same order in this locus. We did not find any counterpart of granzyme M in the fish genome.

## 3. Discussion

Hematopoietic serine proteases perform a number of important functions in mammalian immunity. Granzyme B, which is an enzyme found in cytotoxic T cells and NK cells, induces apoptosis in target cells via the cleavage of caspases [1,37,38]. Mast cells use the chymases to regulate blood pressure via angiotensin cleavage, to regulate the types of immunity by cleaving the selective sets of cytokines, to combat blood-feeding parasites such as mosquitos and ticks via the cleavage of anti-coagulant proteins and to inactivate toxins from snakes and scorpions [4,39,40,41,42,43,44,45,46,47,48]. Neutrophil proteases help these cells to migrate through tissues to reach the site of infection and cleave the pathogen-associated molecules of the infectious organisms [1]. However, the functions of the corresponding proteases among fishes and their targets are almost completely unknown. Relatively solid evidence for granzyme A/K equivalents in fish is present. However, the function of mammalian granzyme A and K have not been well elucidated [49]. However, they are the most well conserved of all hematopoietic serine proteases and are found in essentially all species from cartilaginous fish to humans, but the major question still remain concerning their primary targets [49]. Major similarities are also found among different tetrapods including amphibians, reptiles, and birds, and mammals concerning cleavage specificities and the cell origin of other hematopoietic serine proteases. However, in ray-finned fishes, we know very little about this large family of proteases. Almost all of the non-granzyme A/K members of the hematopoietic serine proteases form their own branch in the phylogenetic tree, indicating that they have experienced an independent expansion and diversification compared to tetrapods (Figure 2). The structure of the ray-finned fish enzymes are also sufficiently different from the mammalian enzymes, so it is not possible to use the positions of the three residues that form the S1 pocket of the tetrapod enzymes to obtain information concerning the primary specificity of the fish proteases [1,2].

What is known about hematopoietic serine proteases in fishes? Granzyme A from the catfish has been previously shown to have cytolytic activity [23]. However, caution around this interpretation may be needed for the human enzyme, despite the reported apoptotic activity, as when more physiological amounts of the enzyme are used, this activity is lost [49,50,51]. A more detailed analysis of this phenomenon may be needed before definitive conclusions can be drawn. A second catfish enzyme, catfish granzyme like-I, has recently been shown to cleave a sequence in catfish caspase 6 that corresponds to the region in human caspase 3 that is cleaved by human granzyme B [17]. However, even here, no direct proof is yet available to validate if this cleavage occurs in vivo. The analysis has been hampered by difficulties in producing recombinant catfish caspase 6, which is needed to enable cleavage analysis with catfish granzyme-like I. However, this enzyme is expressed by catfish NK-like cells, indicating that it may be involved in the apoptosis induction of cells infected with intracellular parasites. The detailed information concerning the cleavage specificities of three additional fish hematopoietic serine proteases is now available, i.e., for catfish granzyme-like II, which is also expressed by the same NK-like cells, and the two enzymes from the spotted gar presented here [32]. The catfish granzyme-like II is a highly specific elastase cleaving after Ala with a strong preference for several basic amino acids, primarily Arg, present at the N-terminal of the cleavage site [32]. To our knowledge, no elastase is expressed by human NK cells, which is why the function of this enzyme may turn out to be a fish-specific immune function. The two gar enzymes are, based on their tissue location and cell shape, expressed in what appears to be NK-like cells or cytotoxic T cells. However, it is not yet clear in what cell these two proteases are expressed due to the lack of suitable reagents to determine the actual cellular origin in gar. Their specificities currently provide little clues to their function. Both enzymes are highly specific. One is a Tyr-specific chymase and the second enzyme a Leu- specific Leu-ase, similar to the mast cell chymases of rabbit and guinea pig, but both appears to be expressed by a different cell type than these latter proteases (Figure 4). Neither chymases nor Leu-ases seem to be expressed by NK or cytotoxic T cells in mammals. The screening of the total proteome with the consensus sequences for these two gar proteases did not result in any clear candidates for their potential immune functions. A relatively broad panel of different potential targets were obtained for both enzymes with no clearly identified immune candidate. A more in-depth analysis involving the direct cleavage of a cell extract and possibly the 2-D gel analysis of cleavage products could be informative. An alternative explanation could be that these enzymes are not aimed at cleaving host proteins but instead pathogen-derived proteins such as bacterial toxins.

The analysis of the intestinal region and the head kidney resulted in an interesting finding concerning the immune organs in this branch of fish evolution. Clear indications for Peyer’s patch-like structures in the intestinal region were observed. Here, the expression of the two gar enzymes was seen, which did not occur in other regions of the fish, including the head kidney, indicating that the proteases are expressed in cells that appear at these sites of immune activity in the intestinal region and not during the early expansion of cells in the head kidney.

A clear separation of regions where red blood cells are formed was seen from the gar head kidney sections and other regions within the same organ where most likely the majority of other hematopoietic cells are residing and was an interesting finding that differs from the organization of human bone marrow. The clusters of red blood cells in the regions of other hematopoietic cells may be due to a rich blood supply to support the rapidly expanding cell populations in need of oxygen and nutrient supply.

The analyses of the genomic organizations of the two gar enzymes also provided insights into the evolution of these proteases. Both of the gar enzymes were located in the met-ase locus (Figure 9). This locus is known in mammals to encode the majority of neutrophil proteases including N-elastase, proteinase 3, azurocidin, and neutrophil protease 4, but also the NK-cell-expressed granzyme M and complement factor D. We also observed that independent amplifications had occurred in mammals compared to gar and sterlet sturgeon (Figure 9). In mammals, it is most likely that the complement factor D is the origin of the majority of the neutrophil proteases, including N-elastase, proteinase 3, and azurocidin, whereas in the gar, it is most likely the PRSS57, encoding NSP4, which has been duplicated, forming a small subfamily of six different proteases in the gar, two in the sturgeon, and three in the bichir (Figure 9). These three fish species are representatives of non-teleost branches of the fish evolutionary tree, indicating that the gene duplications of PRSS57 are early expansions in the met-ase locus in fish [48]. In this regard, the relation and differences between the teleost catfish and zebrafish, and the non-teleost gar, sterlet, and bichir are also interesting. The one or two DDN-like enzymes in one end of the locus found in all three non-teleost fish branches including gar, sterlet, and bichir have been lost in both catfish and zebrafish (Figure 9). The three genes that most likely originate from the gene duplications of PRSS57 are still present in the teleost catfish and zebrafish (Figure 9). Both the catfish granzyme-like II and gar granzyme G have been identified as the first genes of these PRSS57-duplicated genes; however, they have very different cleavage specificities, where catfish granzyme-like II is an Ala-specific elastase and the gar enzyme is a Tyr-specific chymase. Currently, there is no information concerning the other catfish and gar enzymes in this cluster, except for catfish granzyme-like I, which is a met-ase. There are striking similarities between catfish and zebrafish as both catfish granzyme-like I and zebrafish AE-like, as well as catfish granzyme-like II and zebrafish SPA, have very similar specificities [32]. Future studies will hopefully be able to distinguish the similarities and differences in specificities between these enzymes in non-teleost compared to teleost fishes. Such information may guide us toward the key primary immune-related targets for these enzymes during fish evolution and how they relate to their mammalian counterparts.

## 4. Materials and Methods

### 4.1. Phylogenetic Analyses

The phylogenetic analyses, aimed at determining the relationship between the two gar enzymes and other hematopoietic serine proteases from fish, were performed essentially as described in a previous publication using the same strategy and sequences [2]. Sequences relating to the two gar enzymes were systematically uncovered using BLASTp searching of all animal NCBI databases. The mature two gar enzymes were used as the query sequence, and all novel derived sequences were analyzed using the multiple alignment programme MAFFT with G-INS-i strategy and default parameters to determine whether they belonged to the serine protease family. To visualize the relationship between the two gar enzymes with those from other species, a phylogenetic tree using the Bayesian interference of phylogeny algorithm with posterior probabilities in the MRBAYES program was constructed and viewed in FigTree (v1.4). The amino acid sequences for the mature proteins of serine proteases, branching with the two gar enzymes, were aligned using MAFFT.

### 4.2. Production and Purification of Recombinant Gar Gzm-G and Gar DDN1

The gar granzyme sequences (GenBank accession numbers: Gar Gzm-G-like (XP_015220993) and Gar DDN1-like (XP_006640095)) were designed and ordered from GenScript (Piscataway, NJ, USA). The synthesized construct was cloned in the pU57 cloning vector, containing EcoRI and XhoI sites. The gar granzyme sequences were subsequently transferred to a pCEP-Pu2 vector and used for expression in mammalian cells [33]. The enzymes were produced as inactive recombinant proteins, with an N-terminal His_6_-tag followed by an enterokinase (EK) site. HEK 293 cells were grown to 70% confluency in a 25 cm^3^ tissue culture flask (BD VWR) with Dulbecco’s Modified Eagles Medium (DMEM) (GlutaMAX, Invitrogen, Carlsbad, CA, USA) supplemented with 5% fetal bovine serum (FBS) and 50 µg/mL gentamicin. Following transfection with lipofectamine (Invitrogen, Carlsbad, CA, USA), using approximately 25 µg of the gar constructs in pCEP-Pu2, puromycin was added to the DMEM (0.5 µg/mL) to select for cells which had been taken up the DNA. Heparin (5 µg/mL) was added to the culture medium to enhance the recovery of secreted protein. Cells were expanded, and the conditioned media was collected.

After collecting a sufficient quantity of media, in general around 750 mL, the conditioned media was filtered (Munktell 00H 150 mm, Falun, Sweden), and 500 µL nickel nitrilotriacetic acid (Ni-NTA) beads were added (Qiagen, Hilden, Germany) to purify the recombinant enzymes. The media with Ni-NTA beads were rotated for 45 min at 4 °C. Subsequently, the Ni-NTA beads were collected with centrifugation and transferred to a column containing a glass filter (Sartorius, Goettingen, Germany). To remove the unbound protein, the columns were washed with PBS tween 0.05% + 10 mM imidazole + 1 M NaCl. Following this wash, the recombinant protein was eluted in PBS tween 0.05% + 100 mM imidazole fractions. The first fraction volume was half the Ni-NTA bead width (200 µL), and further fractions were eluted with a full bead width (400 µL). Individual fractions were run on SDS-PAGE gel, and their concentrations were estimated from a bovine serum albumin standard (BSA), and the most concentrated were pooled and kept at 4 °C. The yield in production differed significantly between the two proteases. We estimate that the yield for gar DDN1 was approximately 1 mg/L conditioned media, whereas the yield for gar gzm-G was only a tenth of that or approximately 100 µg/L.

### 4.3. Activation of Recombinant Gar Granzymes

To activate the enzymes, the initial concentrations of the gar enzymes were first determined using SDS-PAGE, and the level of EK (Roche, Mannheim, Germany) was adjusted for the activation of the enzymes. To 70 µL of the eluted recombinant enzyme, we added 1 µL EK and incubated for 3 h at 37 °C to activate the enzyme. The activated fractions were stored at 4 °C until use.

### 4.4. Substrate Phage Display

A T7 phage library containing 5 × 10^7^ variants was used for the phage display analysis, to determine the extended cleavage specificity of the two gar enzymes. In this library, each phage displays a unique nine amino acid sequence. The nine-amino acid region had been inserted into the C-terminal of the capsid 10 protein, followed by a His_6_-tag. A 125 µL volume of Ni-NTA agarose beads via their His_6_-tags was gently rotated for 1 h at 4 °C (Qiagen, Hilden Germany). To these 125 µL Ni-NTA, we estimate that approximately 10^9^ plaque-forming units (pfu) were bound. Unbound phages were removed by washing ten times with PBS tween 0.05% + 1 M NaCl, followed by two washes with PBS. The beads were re-suspended in 375 µL PBS, and approximately 250 ng of the recombinant gar enzyme was added. This reaction was incubated overnight or for approximately 16 h at 37 °C with gentle rotation. The enzyme cleavage causes the susceptible phages to detach from the Ni-NTA beads. From the supernatant, released phages were recovered after centrifugation, and thirty µL of which was used in a plaque assay to determine the number of released phages. Briefly, tenfold serial dilutions were made, mixed with (*E. coli*) BLT5615 (for the propagation and visualization of plaques on a bacterial lawn), plated on LA-Amp (50 µg/mL) plates, incubated for 2.5 h at 37 °C, and then counted. The remaining supernatant was added to 10 mL BLT5615 bacteria (OD_600_ 0.5), and the culture was incubated at 37 °C for approximately 75 min until the culture was cleared by phage lysis for phage expansion. From this, 1.5 mL was centrifuged to remove bacterial debris, and 800 µL of this volume was transferred to a microcentrifuge tube containing 100 µL PBS and 100 µL 5 M NaCl. This solution was bound to 125 µL Ni-NTA beads directly after centrifugation to start the next selection cycle. The complete process was repeated a further 6 times, constituting 7 selection rounds. Individual plaques were isolated from the final selection round in 100 µL phage buffer before vortexing for 30 min and stored at 4 °C. The random nine-amino acid regions contained in these phages were amplified using PCR (T7Select primers, Novagen, Sacramento, CA, USA) and sequenced by Eurofins (Sequencing centre, Cologne, Germany). The resulting sequences were translated using CLC viewer and aligned using Adobe Illustrator. A parallel control reaction without enzyme (only PBS) was also run under the same conditions, and plaque numbers were compared to the enzyme sample.

### 4.5. Phage Display Sequence Verification Using a Two-Thioredoxin (trx) Approach

In order to verify the phage display data, a recombinant trx system was used. This system has been developed in our laboratory and used multiple times to study other proteases with great success. Here, the random nine-amino-acid-cleaved region determined from the phage display was introduced between two adjacent *E. coli* trx proteins. Originally, a pET21 vector containing a single trx protein was modified to contain a second trx with BamHI and SalI sites in the intervening region. Here, the random region was synthesized as oligonucleotides (Sigma, St. Louis, MO, USA). These oligos were generated to be suitable for the ligation between BamHI and SalI restriction sites of the vector. This resulted in a vector containing a first trx, followed by the random cleavable region, and then a second trx with His_6_-tag (facilitating purification).

To produce the recombinant protein, the construct was expressed in *E. coli* Rosetta gami (Novagen, Sacramento, CA, USA). Ten ml of an overnight culture was added to 90 mL LB + Amp (50 µg/mL) and 500 µL 20% glucose. After approximately 1 h (reaching OD_600_ 0.5), 100 mM isopropyl β-D-1-thiogalactopyranoside (IPTG) was added, and the culture was placed on a shaker (with moderate shaking) at 37 °C for 3 h. The bacterial cells of the culture were pelleted using centrifugation at 10,000 rpm for 3 min, and the supernatant was discarded. The pellet containing the bacteria with the expressed recombinant protein was washed in 10 mL PBS tween 0.05%, centrifuged, and pelleted again, followed by resuspension in 1/100th starting volume (i.e., 1 mL) PBS. To obtain the intracellularly expressed protein, the resuspended pellet was sonicated for 6 × 30 s on ice. The supernatant was transferred to a new microcentrifuge tube after centrifugation at 10,000 rpm for 10 min at 4 °C. To purify the protein, 125 µL Ni-NTA beads were added and incubated for 45 min at 4 °C with gentle agitation (Qiagen, Hilden Germany). The solution with Ni-NTA beads was then transferred to a 2 mL column (Terumo, Leuven, Belgium) containing a glass filter (Sartorius, Goettingen, Germany). The column was then first washed with 3 × 2 mL and 2 × 1 mL PBS tween 0.05% + 10 mM imidazole. To elute the protein, fractions were collected after passing through PBS tween 0.05% + 100 mM imidazole. The first fraction volume was only half the Ni-NTA bead volume (75 µL), and further fractions were eluted with a full bead volume (150 µL). Individual fractions were run on SDS-PAGE gel, and their concentrations were estimated from a BSA standard (using a Bradford assay (Bio-rad, CA, USA)). The most concentrated fractions were then pooled and kept at 4 °C. until use in the assay.

For cleavage analysis, approximately 250 ng of recombinant gar enzyme was added to 20 µg of the pooled 2-trx protein (containing different sequences based on the phage display data), and the aliquots of 5 µg were removed after 0, 15, 45, and 150 min of enzyme addition. The reactions were run at room temperature, and the time point aliquots analyzed on SDS-PAGE gel under denaturing conditions using pre-cast 4–12% Bis-Tris gels (Invitrogen, Carlsbad, CA, USA) and 1x MES buffer (Invitrogen, Carlsbad, CA, USA). Gels were stained with Coomassie colloidal solution to visualize the protein bands [52].

### 4.6. Screening for Potential In Vivo Substrates

The derived consensus sequences for the two gar proteases (Ala-Val-Phe-Tyr-Ser-Leu) and (Arg-Arg-Leu-Pro-Leu-Leu), respectively, were used for the screening of the potential in vivo substrates using a standard protein Basic Local Alignment Search Tool (BLASTp).

### 4.7. Spotted Gar Husbandry and Tissue Sampling

Spotted gars were raised up to their juvenile stages from fertilized eggs obtained from the hormone-induce spawns of wild-caught broodstock from bayous near Thibodaux, Louisiana. Five unsexed individuals were euthanized at the age of 4 months (average total length: 15.9 cm) using MS-222 (250–500 mg/L water) and fixed overnight in 4% Paraformaldehype/PBS. All animal studies were approved by the Michigan State University Institutional Animal Care and Use Committee (PROTO202200298).

### 4.8. Immunohistochemistry

Approximately 100 µg of the recombinant enzymes was mixed with Freund’s complete adjuvant and injected in one male 20-week Sprague Dawley rat for each enzyme. Three weeks after the first injection, the rats were boosted with 20 µg of the recombinant protein in Freund’s incomplete adjuvant. After 6 weeks, another booster dose was given with 20 µg of recombinant protein in Freund’s incomplete adjuvant. Two weeks after the last booster injection, the rats were tail bleed to obtain approximately 1 mL of blood. The blood was left overnight to coagulate and thereafter spun at 3000 rpm in an Eppendorf centrifuge, and the serum was transferred to a new tube and centrifuged a second time to remove the remaining red blood cells. The serum was then aliquoted and kept at −80 °C, until it was used for immunohistochemical staining.

The fish were preserved in a formalin solution for several days. Following preservation, the tissue was decalcified using 0.5 M ethylenediaminetetraacetic acid at pH 8.0 (Sigma Aldrich, Saint Louis, MO, USA) for a duration of one month, during which the solution was replaced on a weekly basis. Subsequently, the tissue was immersed in a 30% sucrose solution in 1x PBS for 72 h before sectioning. A transverse incision was then performed at the tissue’s midpoint, extracting a section of approximately 1 cm^2^ around the intestinal cavity with the use of a scalpel. Then, the dissected tissue was embedded in the optical cutting compound (OCT, VWR, Radnor, PA, USA). Sections of 15 µm thickness were obtained using a cryostat and mounted onto SuperFrost slides (VWR, Radnor, PA, USA).

For immunostaining, the tissue sections were initially blocked for a period of 1 h at room temperature with 1x protein-free blocking buffer (Pierce solution, Thermo Fisher, Waltham, MA, USA) containing 0.3% Triton X-100 in 0.1 M PBS. Staining with the rat anti gar-DDN1 and anti-GzmG (1:100) primary antibody was performed overnight at 4 °C. After the primary antibody incubation, the tissue sections were triple-washed with a PBS solution. Subsequently, the sections were incubated with a donkey anti-rat secondary antibody conjugated to Alexa 488 (1:500, Thermo Fisher, Waltham, MA, USA) for 2 h at room temperature. Nuclei were visualized using DAPI (4′,6-diamidino-2-phenylindole) (Thermo Fisher, Waltham, MA, USA) and mounted using Fluoromount G (Thermo Fisher, Waltham, MA, USA). Pictures were taken using an epifluorescence Nikon microscope (Nikon Eclipse 90i Upright).

## Figures and Tables

**Figure 1 ijms-25-01669-f001:**
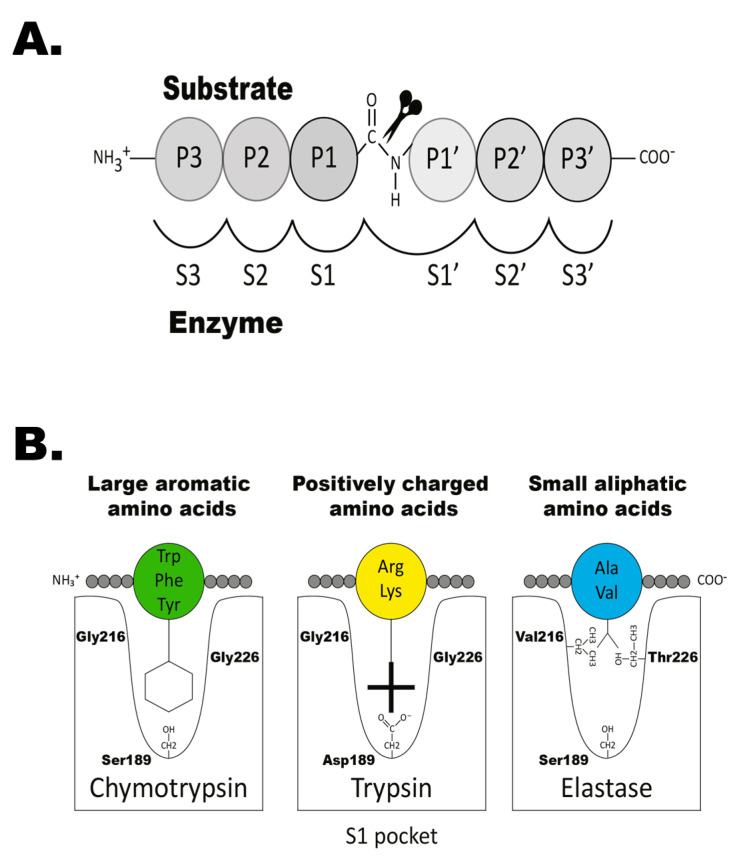
Nomenclature of the amino acids surrounding the cleavage site and the amino acids forming the active site pocket. In panel (**A**)**,** the nomenclature of the amino acids surrounding the cleaved peptide bond is shown. The amino acid N-terminals from the cleaved bond are termed as P1 (where cleavage occurs, depicted by scissors), P2, P3, etc. Amino acid C-terminals of the cleaved bond are termed as P1′ (adjacent to P1), P2′, P3′, etc. The corresponding interacting sub-sites in the enzyme are denoted with S. In panel (**B**), the three amino acids forming the active site pocket (S1 pocket) are shown. These three residues correspond to positions 189, 216, and 226 in bovine pancreatic chymotrypsinogen and have been found to determine the primary specificity of the enzyme as either chymotrypsin-, trypsin-, or elastase-like specificity [31]. The preferred amino acids in the P1 position of the corresponding substrates are illustrated in green, yellow, and blue, respectively. Figure reproduced from [32].

**Figure 2 ijms-25-01669-f002:**
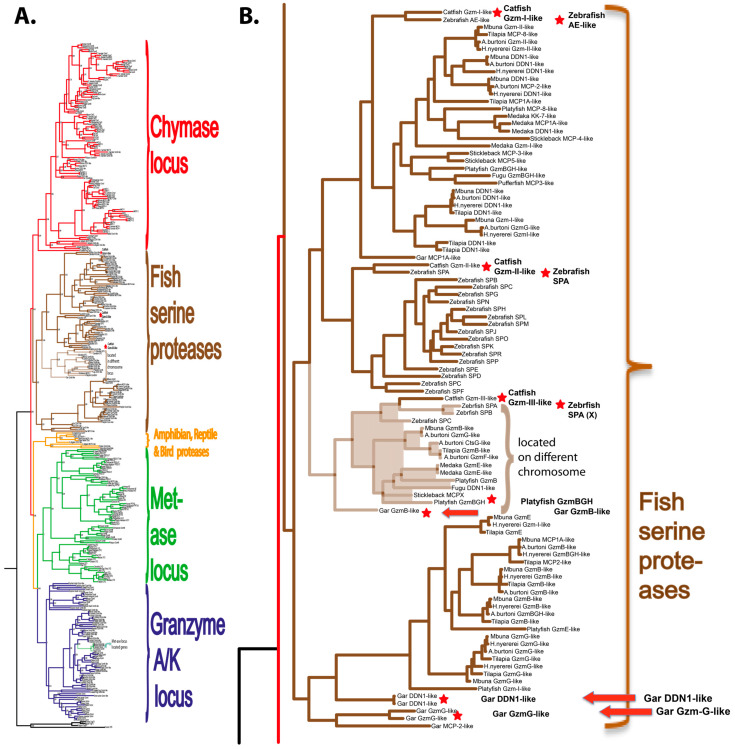
Phylogenetic relationship between gar granzyme G, gar DDN1, and other hematopoietic serine proteases. All sequences were run in the multiple alignment programme, MAFFT, to verify if they belonged to the serine protease family. The tree was constructed using MRBAYES with a Bayesian interference of phylogeny algorithm (with posterior probabilities), opened with FigTree (v1.4) and annotated in Adobe illustrator (CS5). Panel (**A**) shows the entire analysis involving a total of 368 vertebrate serine protease sequences. Panel (**B**) shows an enlargement of the branch of the major tree, where the majority of the fish proteases are found, except the granzyme A/K-related fish proteases. The proteases of particular interest for this study are marked with red arrows. All genes for proteins that were produced as recombinant proteins are marked with red stars. Figure adapted from [32].

**Figure 3 ijms-25-01669-f003:**
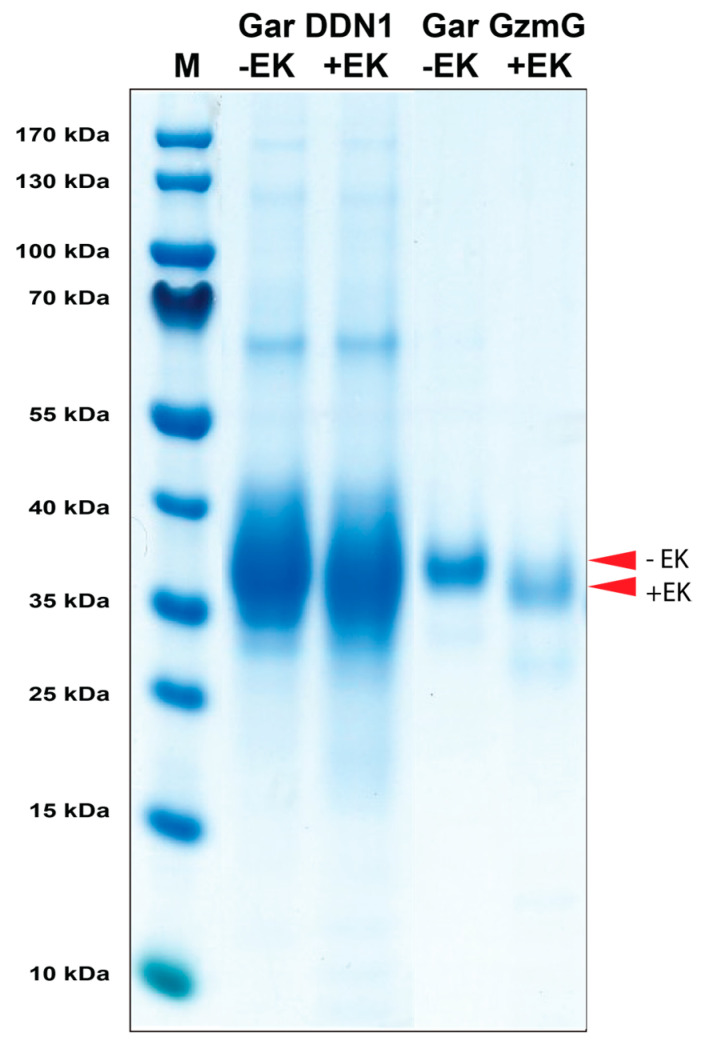
Recombinant gar granzyme G and gar DDN1. The enzymes were produced as an inactive protein (**left lane**) in HEK293 cells with an N-terminal His_6_-tag and enterokinase (EK) site, facilitating purification and activation, respectively. The addition of EK cleaves the N-terminal sites, resulting in an active enzyme and a subsequent drop in size (**right lane**). The enzymes were run on a 4–12% pre-cast SDS-PAGE gel (Invitrogen, Carlsbad, CA, USA) and stained with colloidal Coomassie brilliant blue.

**Figure 4 ijms-25-01669-f004:**
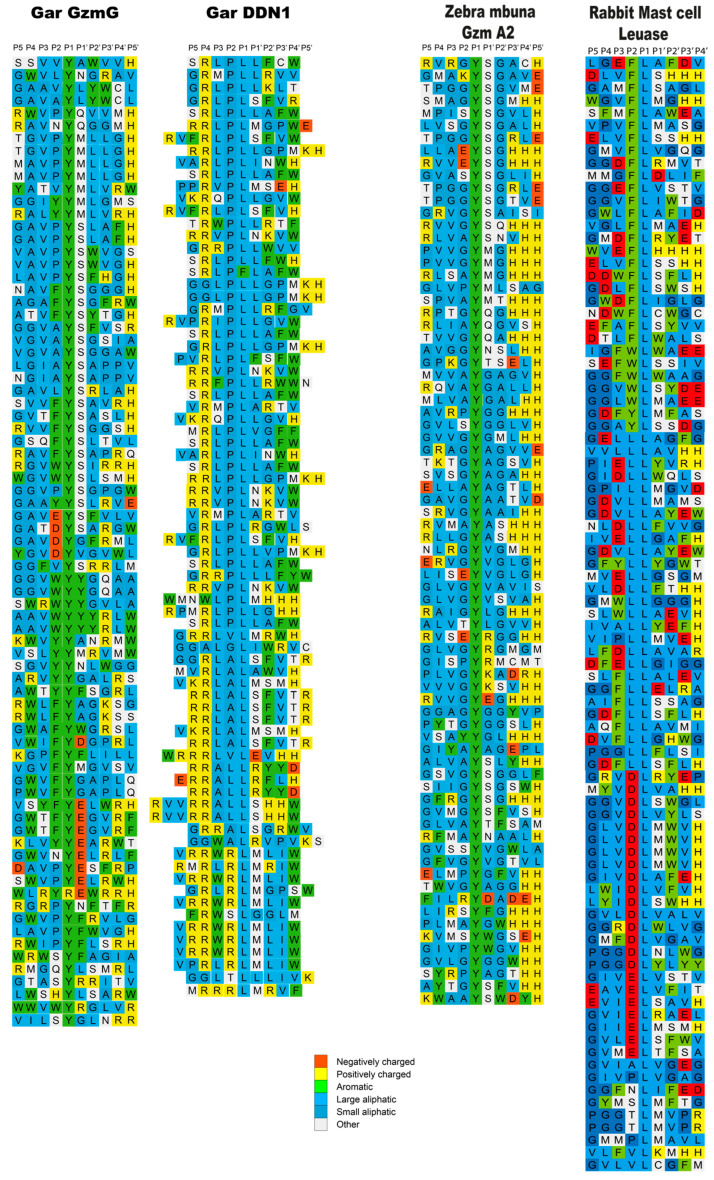
Substrate phage display sequences. The phage display-derived sequences from the analysis of both gar granzyme G and gar DDN1 are displayed in the left part of the figure. Each line represents a separate phage-derived sequence. For comparison, two previous phage display analyses are included, with specificities similar to what was observed for gar granzyme G and gar DDN1. Gar granzyme G, which has a P1 preference for Tyr, shows similarities to a recently analyzed *Zebra mbuna* enzyme named granzyme A2 [34]. The gar DDN1 shows some similarities to rabbit and guinea pig Leu-ases, as they share the same P1 preference [35].

**Figure 5 ijms-25-01669-f005:**
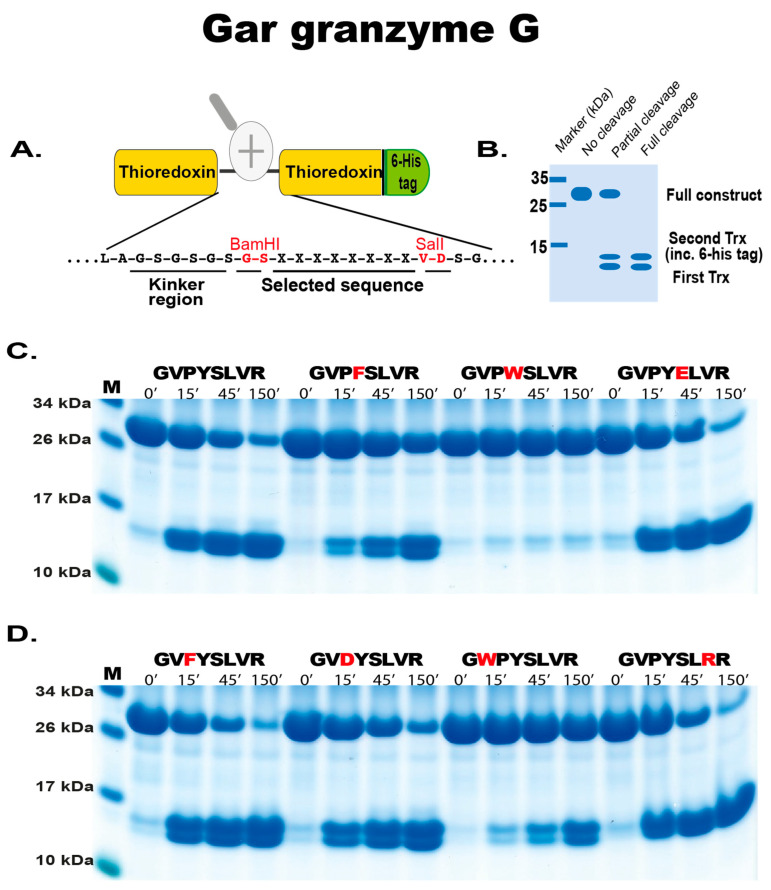
Verification of phage display sequences using the 2xTrx system. A number of phage display-derived sequences and variants of these sequences were added in between two adjacent trx proteins (panel (**A**), adapted from [32]), expressed in *E. coli* and subjected to gar granzyme G (panels (**C**,**D**)). The results were run on pre-cast 4–12% SDS-PAGE gels (Invitrogen, Carlsbad, CA, USA). Hypothetical cleavage is shown (panel (**B**)) to highlight the possible cleavage patterns. The individual lanes represent various time points after the addition of the enzyme, in minutes.

**Figure 6 ijms-25-01669-f006:**
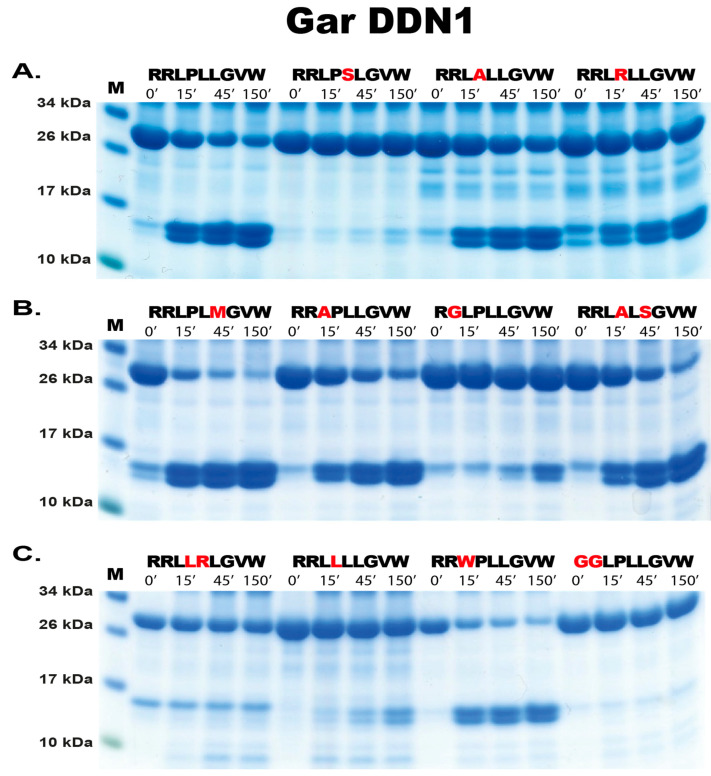
Verification of phage display sequences of gar DDN1 using the 2xTrx system. A number of phage display-derived sequences and variants of these sequences were added in between two adjacent trx proteins, were expressed in *E. coli*, and were subjected to gar DDN1 (panels (**A**–**C**)). The results were run on pre-cast 4–12% SDS-PAGE gels (Invitrogen, Carlsbad, CA, USA). The individual lanes represent various time points after the addition of the enzyme, in minutes.

**Figure 7 ijms-25-01669-f007:**
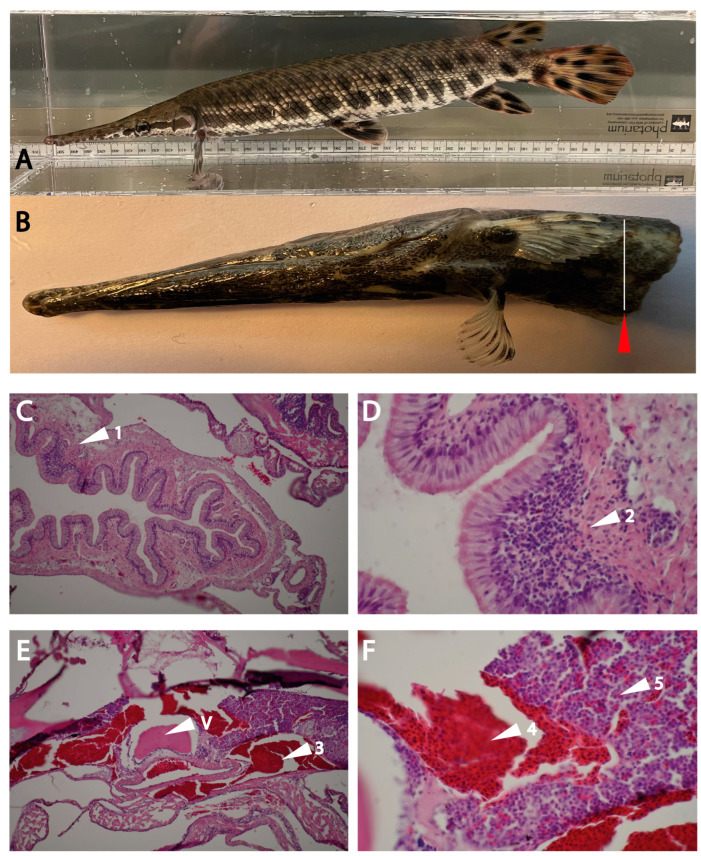
Histological analysis of the gar tissues. Panel (**A**) shows a picture of a gar. Panel (**B**) shows a picture of the juvenile gar used for the histological analysis of head kidney and intestinal regions for immune cells. The section of the animal that was used for analysis is marked with a white line and a red arrowhead. Panels (**C**–**F**) show hematoxylin–eosin-stained tissue sections. Panel (**C**) shows primarily the intestinal region with a white arrowhead (1) pointing at a Peyer’s patch-like structure in the intestinal wall. Panel (**D**) shows an enlargement of this Peyer’s patch region (arrowhead 2). Panels (**E**,**F**) show the head kidney region with arrowheads, marking the pure red-blood-cell-rich regions of the head kidney (arrowhead 3). Arrowhead V shows the vertebra. In panel (**E**), arrowhead number 4 shows an enlargement of this red blood region, and arrowhead 5 shows the immune-cell-rich region of the head kidney.

**Figure 8 ijms-25-01669-f008:**
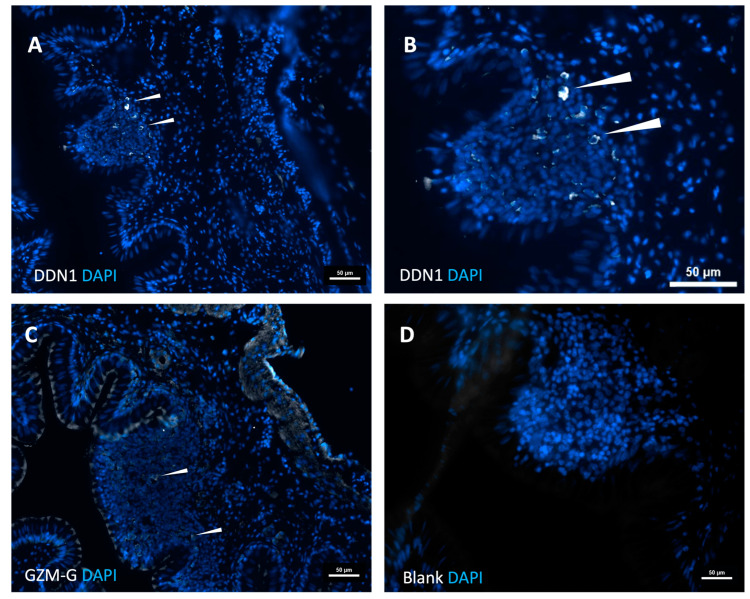
Staining using DDN1 and GZM-G antibodies. In panel (**A**), a representative section of DDN1 staining is shown, and in (**B**), a magnification of this section is shown for clarity. In panel (**C**), a representative section of the staining of gzm-G antibody is shown, and in panel (**D**), the primary antibody control of fish tissue is shown. Nuclei were stained using DAPI in blue.

**Figure 9 ijms-25-01669-f009:**
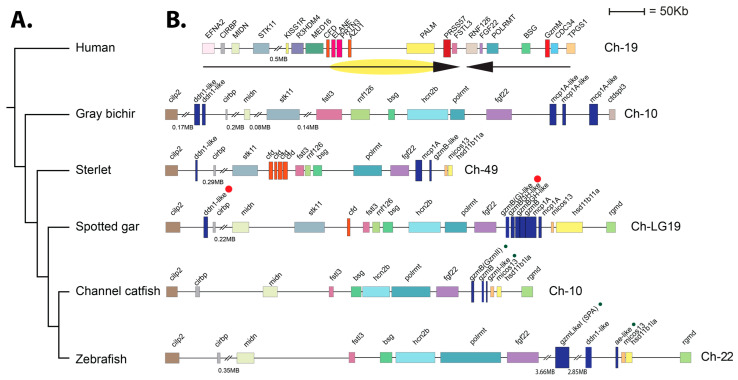
Genomic loci encoding fish hematopoietic serine proteases. The locus encoding both enzymes is encoded in the ray-finned fish met-ase locus. This locus also exists in mammals, including humans, with a slightly different organization. The human locus has most likely experienced an inversion and an internal expansion, marked with a yellow oval just below the human locus. Serine protease genes have a double height for visual identification. Red dots mark the two enzymes analyzed in this communication, and blue dots mark enzymes that were previously analyzed. Green dots mark the two catfish proteases analyzed in [32] and the two zebra fish proteases that show similarities to the cafish proteases discussed in [32] and in the discussion of this manuscript. Panel (**A**) shows a tree of fish evolution based on both genetic and morphological information, where bichir is the first to branch out from the common fish branch followed by starlet and later the gar and finally the teleosts [24,36]. Panel (**B**) shows the loci of the different fish species specified in panel (**A**).

## Data Availability

All data of importance for the study is available in the article.

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
