# Peer review of "Extended Cleavage Specificity of two Hematopoietic Serine Proteases from a Ray-Finned Fish, the Spotted Gar (Lepisosteus oculatus)"

_ijms, 2024, doi:10.3390/ijms25031669_

Round 1
Reviewer 1 Report
Comments and Suggestions for Authors
The spotted gar is widely used with regard to evolution and also to understand basic aspects of teleosts, so the focus of the study was simply correct and very interesting, using different techniques to study hematopoietic serine proteases. Therefore, I offer my opinion with some minor revisions.
The draft (especially in the introduction) could bring a little more of the evolutionary aspect, such as the third Whole Genome Duplication, specifically in teleostelis.
L 116 - I believe that the configuration in figure 2 is very difficult to follow, the tree is very large with low resolution (at least in the draft)
I would like to emphasize that the draft is very well designed and written, therefore I recommend acceptance of it
Author Response
We have now added a section in the introduction describing the different tetraploidizations that has affected the evolution of vertebrates and in particular fishes including the teleost 3R. Four new references to these important events have also been added to this text. The new text is marked in red.
The figure 2 has now been updated to more than double resolution. We have also updated the resolutions of figures 3 and 4.
Reviewer 2 Report
Comments and Suggestions for Authors
The authors wrote about identifying two new serine proteases from ray-finned fish spotted gar (Lepisosteus oculatus). They describes the extended cleavage specificities of two enzymes. I suggest the minor points for clear understanding.
1) Fig.1A : It is not necessary because of too general conception.
2) Please show the specific yield of each product. And show the band with an (red) arrow.: Fig. 3
3) page 164-173 Figure 4. Too many (figure4).
4) Fig 7. Does have your picture of gar? Normally not used in Wikipedia reference pictures.
Best.
Author Response
We know that the figure 1 A is a general description. However, from earlier experience we know that by adding this type of explanatory figures we increase the understanding of the text to a broader audience. For everyone working in the field of protease function this is textbook knowledge. However, from experience with trying to explain our studies for a broader immunological audience we have experienced that a large fraction of the audience has difficult to understand the explanations or the results from phage display and the recombinant substrates and as this article is aimed for a broader audience working in the fields of evolution, immunology and biochemistry why we would like to keep this simple explanatory figure.
We have now added description of the recombinant proteases and also added red arrows for pre and post enterokinase samples in figure 3.
We have now changed and reduced the text to figure legend 4.
We have now added a new inhouse figure of a spotted gar in figure 7.
The changes and additions in the text is marked in red.